# New Therapeutic Strategies in Retinal Vascular Diseases: A Lipid Target, Phosphatidylserine, and Annexin A5—A Future Theranostic Pairing in Ophthalmology

**DOI:** 10.3390/ph17080979

**Published:** 2024-07-24

**Authors:** Anna Frostegård, Anders Haegerstrand

**Affiliations:** 1Annexin Pharmaceuticals AB, Kammakargatan 48, S-111 60 Stockholm, Sweden; 2Unit of Immunology and Chronic Disease, IMM, Karolinska Institute, S-171 77 Stockholm, Sweden

**Keywords:** Annexin A5, anti-VEGF, retinal vein occlusion, phosphatidylserine

## Abstract

Despite progress in the management of patients with retinal vascular and degenerative diseases, there is still an unmet clinical need for safe and effective therapeutic options with novel mechanisms of action. Recent mechanistic insights into the pathogenesis of retinal diseases with a prominent vascular component, such as retinal vein occlusion (RVO), diabetic retinopathy (DR) and wet age-related macular degeneration (AMD), may open up new treatment paradigms that reach beyond the inhibition of vascular endothelial growth factor (VEGF). Phosphatidylserine (PS) is a novel lipid target that is linked to the pathophysiology of several human diseases, including retinal diseases. PS acts upstream of VEGF and complement signaling pathways. Annexin A5 is a protein that targets PS and inhibits PS signaling. This review explores the current understanding of the potential roles of PS as a target and Annexin A5 as a therapeutic. The clinical development status of Annexin A5 as a therapeutic and the potential utility of PS-Annexin A5 as a theranostic pairing in retinal vascular conditions in particular is described.

## 1. Background

Long-term visual impairment and blindness are detrimental life-affecting outcomes of vascular retinal disease. With the global growth of the aging population, the impacts of diabetic retinopathy (DR), age-related macular degeneration (AMD), and retinal vein occlusion (RVO) are expected to become even more significant [1,2].

The current mainstay of vascular and degenerative retinal disease treatment is a monotherapy with intravitreally administered anti-vascular growth factor agents (anti-VEGFs). This treatment paradigm is aimed at mitigating the consequences of VEGF signaling, mainly macular edema and angiogenesis; however, it has limited potential to modify the underlying key triggers of VEGF production, such as retinal ischemia [3]. Despite the widespread use of anti-VEGFs in real-world settings, a durable improvement in vision is not observed across all patients, and the necessity of repeated “in-office” monitoring to optimize the treatment schedule poses a high burden on patients and health systems [4].

Novel therapeutics addressing inflammatory components via the inhibition of complement pathways [5] or the simultaneous targeting of VEGF and angiopoietin 2 [6], both of which are involved in vascular retinal destabilization, have only recently been approved with the hope of providing effective and more durable options for patients. In addition, new formulations enabling a higher intravitreal dosing of anti-VEGFs [7] are available. However, there is still an unmet need for safe and effective medicines with novel mechanisms of action for the management of patients with vascular retinal diseases [4].

Phosphatidylserine (PS) is a lipid constituent of cell membranes that is strictly secluded inside the cell and is externalized in exceptional, mostly pathologic, circumstances [8]. PS has been postulated to be a pharmacologic target in a number of diseases, including vascular and degenerative retinal diseases, such as DR [9], wet AMD [10], and RVO [11]. 

Annexin A5 is a human endogenous protein that binds PS specifically and with high affinity [12]. Treatment with Annexin A5 counteracts the downstream effects of PS externalization, as demonstrated in multiple pre-clinical studies [13,14,15,16,17,18]. On the basis of in vitro, ex vivo, and in vivo pharmacology data, intravenously administered Annexin A5 is expected to rapidly bind to PS-exposing cell membranes. By building a shield over PS and by neutralizing the overexposure of PS, Annexin A5 is postulated to act as an anti-adherent [11], anti-inflammatory [19], on-site anti-thrombotic (as opposed to a systemic anti-coagulant) [20], and vascular-stabilizing cell membrane repair agent [21].

In addition to retinal diseases, the therapeutic use of Annexin A5 to target PS has been proposed in other ophthalmic diseases [22,23], cancer [17], sepsis [18,24], viral infections including COVID-19 [25,26], inflammatory bowel disease [27], osteoporosis [28], Alzheimer’s disease [13], vaso-occlusive crisis in sickle cell disease [29,30], diabetic skin wounds [16], renal ischemia reperfusion damage [15], and cardiovascular disease [14]. 

Recombinant human Annexin A5 proteins (rhANXA5) are currently in clinical development as a potential therapeutic in RVO (ANXV, Phase 2a Safety and Proof of Concept Study, ClinicalTrials.gov Identifier: NCT05532735; Annexin Pharmaceuticals, Stockholm, Sweden) and in sepsis (SY-005; Phase 2a, ClinicalTrials.gov Identifier: NCT04898322; Yabao Pharmaceutical Group Co., Ltd., Yuncheng, China). SY-005 was also investigated in patients with severe COVID-19 (ClinicalTrials.gov Identifier: NCT04748757) [31,32]. The potential clinical utility of fluorescently labeled Annexin A5 as a molecular diagnostic agent targeting PS on stressed/dying cells has been evaluated in dry AMD and glaucoma [33,34]. Multiple molecular imaging studies have used technetium-99m (99mTc)-labeled Annexin A5 in patients with cancer [35] and rheumatic disease [36].

ANXV is an investigational new drug that contains human protein Annexin A5 produced by recombinant techniques in *E. coli*. It is currently in clinical development by Annexin Pharmaceuticals (Stockholm, Sweden). ANXV has been investigated in pre-clinical settings, having completed one Phase 1 First in Human clinical trial in male healthy volunteers (ClinicalTrials.gov Identifier: NCT04850339), where it was well tolerated across the evaluated dose ranges. The first study with ANXV in a patient population, a Proof of Concept/Phase 2a study (NCT05532735), is ongoing in the USA, and enrollments are limited to VEGF-treatment naïve patients with recently diagnosed RVO. ANXV treatment is initiated within 28 days post symptom onset, and the mechanism of action suggests that ANXV may be effective in reducing the size of occlusion in the retinal vein and act against the immediate pathologic consequences thereof, such as retinal ischemia, as well as against inflammation and retinal damage. Hence, ANXV might be a promising novel agent for vascular and degenerative retinal diseases, targeting pathways that are upstream of VEGF and complementing signaling.

## 2. Retinal Vein Occlusion and Other Vascular Retinal Diseases

The global ageing population is rapidly growing, with the population aged 65 and above expected to rise from 10% in 2022 to 16% in 2050 [37]. Accordingly, the impact of vascular and degenerative eye diseases that are more prevalent in the ageing population is expected to become even more significant with time [1,2]

RVO is an occlusion of the central or the branch retinal vein and is the second most common sight-threatening retinal vascular disorder after diabetic retinopathy [38]. Globally, the prevalence of RVO is reported to be from 16 [38] to over 28 million [39].

RVO is a multifactorial disease with risk factors that include ageing, systemic hypertension, dyslipidemia, smoking and glaucoma [40,41]. The exact pathogenesis is still debated, with the occlusive phenomenon potentially not exhibiting all of the hallmarks of the classic thrombus [42], although enhanced erythrocyte aggregation and blood hyperviscosity have been observed [11,43]. Compression of the vein at arteriovenous crossings might contribute, at least in part, to the etiology of RVO [44]. Both systemic and local inflammation are linked to RVO’s pathogenesis [45,46]. Recent reviews on RVO and novel pharmaceuticals currently in development are available [47,48]. 

Briefly, there are two distinct types of RVO based on the anatomic location of the occluded vein—central (CRVO) or branch (BRVO) [49]. All RVO types are likely to present with a sudden, painless loss of vision and macular edema (ME) in one eye. Vision impairment in RVO is due to a combination of factors where the complications of RVO develop over time, with persistent macular edema leading to VEGF-dependent angiogenesis, inflammation, and irreversible retinal damage. Ischemic CRVO is more likely to lead to severe and/or progressive vision loss, whereas there is more potential for partial spontaneous recovery following non-ischemic CRVO or BRVOs [47]. 

The current strategies focus on minimizing the consequences of RVO to reduce vision loss resulting from macular edema and neovascularization [50]. The treatment options for patients with RVO include intravitreal VEGF inhibition, the use of corticosteroids to reduce inflammation, and laser treatment to prevent neovascularization. No treatments to date for macular edema or RVO have demonstrated reliable direct improvements regarding retinal perfusion or displayed a durable disease-modifying effect. Despite the excellent efficacy of anti-VEGFs in reducing macular edema and the subsequent rapid improvement of visual acuity, in many patients, the efficacy tends to decline when injection frequency is reduced [51], while in others, persistent macular edema is obvious, despite intensive VEGF treatment [52]. The currently available RVO treatments are invasive, chronic, and expensive. There are concerns in terms of both their efficacy and durability in real-world settings as opposed to clinical trial data. There are also safety concerns associated with frequent intravitreal injections ranging from the mild to devastating events such as eye pain [53] to infectious endophthalmitis [54]. New approaches currently being evaluated include increasing the dose of anti-VEGFs to reduce the number of intravitreal (IVT) injections beyond the first 6 months [55]. Recently, a new drug combining a VEGF inhibitor and an angiopoietin-2 (Ang-2) inhibitor was approved for treating macular edema in RVO [56]. 

While these recent approvals can reduce some of the burden on patients, novel targets, novel mechanisms of action, and therapeutic approaches that complement VEGF-based strategies are of high interest and relevance. New mechanisms of action that can be targeted with monotherapy or a combination of therapies that can reduce the high treatment burden of frequent intraocular injections, improve visual outcomes and prevent the risk of progressive vision loss in retinal degenerative and vascular disease need to be explored [48]. 

## 3. PS as a Novel Therapeutic Target in Human Disease

Phosphatidylserine (PS) is an abundant lipid component of all cell membranes, comprising 10–15% of the total phospholipid content [8]. In a healthy state, PS is strictly maintained on the inner leaflet of the cell membrane by the ATP-dependent enzymes flippases [57]. Loss of asymmetric PS membrane distribution occurs when cellular homeostasis is disturbed, with floppases and PS-specific scramblases rapidly orchestrating the externalization of PS to the outer cell membrane leaflet [57].

Externalized PS acts a critical molecule of multiple interrelated processes [58,59] such as immunomodulation, hemostasis and inflammation, while the intracellular functions of PS are less well understood [58]. It is clear that PS is essential for life, since mammalian cells lacking the PS-synthetizing machinery do not survive [60]. 

Under physiological conditions, PS externalization occurs on single cells that are committed to programmed cell death, where it is thought to be irreversible and acts as an “eat me” signal for the phagocytes [61]. In viable cells, transient and reversible PS externalization occurs on stressed cell membranes and early apoptotic cells and is involved in physiologic processes, such as myoblast or osteoclast fusion [62]. 

Under pathologic circumstances, aberrant sustained PS externalization is critical for mediating coagulation, inflammation, cell aggregation, and immunomodulation and is seen as a “fingerprint of tissue damage” [27]. Tumor vascular cells and virus-infected cells lose their capacity to maintain PS asymmetry [25,63]. In many cell types, PS externalization is triggered directly and rapidly via hypoxia, oxidative stress, infection, and other types of cell stress/activation/damage caused by, e.g., advanced glycation end products (AGEs) [64] and amyloid-β (Aβ) peptide [65]. As such, PS signaling occurs upstream of other pathways, including complement factor activation and VEGF synthesis induced by hypoxia. Externalized PS is a trigger able to activate both the alternative and the classical complement pathways [66,67]. There is also evidence of PS signaling axis involvement in VEGF regulation [68]. It is plausible that PS-exposing retinal cells that are transiently stressed by, e.g., hypoxia contribute to the pathology and symptoms experienced by patients suffering from vascular retinal diseases.

As one of the key regulators of the immuno-thrombogenic cascades, PS, exposed on membranes of platelets, erythrocytes, and endothelial cells, provides an essential catalytic surface for the assembly of tenase and pro-thrombinase complexes (intrinsic contact coagulation pathway), thereby augmenting thrombin generation and promoting a hypercoagulable tissue surface [59,69]. The production of active thrombin during this process can in turn propagate inflammation [70]. PS exposure is also essential for the expression of tissue factors and their activity (extrinsic coagulation pathway) [71]. 

PS is a key mediator driving cellular responses to hypoxia. Externalized PS on endothelial cells, platelets, and microparticles acts as a substrate for secretory phospholipase A2 [72], resulting in the subsequent generation of lysophosphatidic acid (LPA), a major mediator of inflammation and reperfusion injury. The downstream detrimental LPA-induced effects include the loss of vascular integrity, platelet aggregation, and macrophage activation [72,73]. Sustained cell-surface exposure of PS is involved in the interplay of danger-associated molecular patterns triggering interleukin 1β (IL-1β)- and interleukin 6 (IL-6)-related pathways, providing a mechanistic link to vascular and tissue inflammation [74]. 

## 4. PS as a Novel Therapeutic Target in Ophthalmology

Aberrant persistent PS externalization is reported to be responsible for the pathogenic adhesion of cells to the endothelium in diabetes, polycythemia vera, and sickle cell disease, where the data demonstrate a striking correlation between erythrocyte PS positivity and their propensity for endothelial adhesion [11,30,75]. In patients with RVO, aberrantly externalized PS on erythrocytes has been identified as a key molecule responsible for the pathologic adherence of erythrocytes to the endothelium [11] and for the increased hypercoagulable state in RVO [43]. In diabetic retinopathy, the quantities of PS on platelets and monocytes are correlated with the aggressiveness of disease and hypercoagulable propensities, being significantly higher in patients with proliferative DR as compared to non-proliferative DR and controls [9].

The percentage of PS-positive circulating cells (as determined by Annexin A5 binding) as well as the number of Annexin A5pos sites per cell as measured ex vivo are increased significantly, e.g., for RVO [11] and sickle cell disease [76,77]. An ex vivo study by Wautier et al. demonstrated that erythrocytes from patients with recently diagnosed RVO have significantly higher capacity to aggregate and adhere to the microvascular endothelium under both static and physiologic flow conditions [11]. In this relatively small study, a significantly higher percentage of erythrocytes exposing PS was found in RVO patients as compared with erythrocytes from normal subjects (*p* < 0.001). Moreover, PS-positive erythrocytes from CRVO patients had a 2.4-fold higher PS target density per cell as compared to healthy controls, and this correlated with erythrocyte adhesion to the endothelium (*p* < 0.001). Currently, it is not known whether such aberrant PS exposure on a subset of RVO erythrocytes is of a transient character and how long after an occlusive event it may last. 

The immunomodulating properties of externalized PS are usually described in relation to apoptosis and cancer, where PS on non-apoptotic cancer cells acts as a potent immunosuppressor by interacting with multiple immunocompetent cells [78]. Cancer cells may constitutively externalize PS at a density of up to 90-fold greater than that of their healthy counterparts when quantified on the surface of individual cells in vitro [79], and PS acts as an inhibitory immune checkpoint within the tumor microenvironment [80].

The externalization of PS is also relevant to the area of extracellular vesicles. Extracellular vesicles of various sizes in the range of 100–1000 nm (microparticles and exosomes; EV) are a relatively new area of interest in disease pathogenesis and as a potential therapeutic target in general as well as in ophthalmology [81]. These vesicles are released from stressed or activated cells, such as endothelial and retinal pigment epithelial cells [82], and are found either circulating in the blood or in various body fluid compartments, including the vitreum. They can cross the blood retinal barrier and contain multiple mRNAs that modulate signal transduction between the cells, participating in, e.g., the regulation of inflammation and angiogenesis. 

The currently available data support a role of EV in the retinal microenvironment, potentially modulating pathophysiology in multiple retinal vascular and degenerative diseases, such as RVO, AMD, DR, and glaucoma [83]. PS-positive EVs derived from systemic circulation in RVO were demonstrated to be responsible for a shortened clotting time with the upregulation of Factor Xa and thrombin formation [43]. Significantly increased levels of PS-positive EV from multiple sources were found in the vitreous of patients with retinal detachment and were independently correlated with vitreous pro-inflammatory cytokines such as monocyte chemoattractant protein-1 (MCP-1) [84]. Interestingly, the authors proposed that EVs represent the missing link between the mechanical stress induced by retinal detachment and the occurrence of subsequent inflammatory and cell activation cascades.

Given the data pointing toward the involvement and relevance of PS for vascular and degenerative retinal diseases, strategies of blocking PS have been investigated in several pre-clinical settings studies. For example, inhibiting PS signaling using antibodies against the plasma protein beta2-Gp1 was successful in reducing choroidal angiogenesis both ex vivo and in vivo [10], whereas blocking PS via direct Annexin A5 binding on erythrocytes from RVO patients ex vivo in flow conditions resulted in up to a 70% reduction in their adhesion to the endothelium [11]. 

Since PS externalization is inversely correlated with intracellular levels of ATP, the most dramatic widespread PS externalization is expected to occur in a model of hemorrhagic shock, also representing the most extreme situation of hypoxia. Depletion of ATP sources incapacitates PS membrane asymmetry and is accompanied by micro-thrombosis, vascular leakage, and cell/tissue injury [85]. The therapeutic strategy of blocking PS in models of hemorrhagic shock was shown to prevent renal dysfunction and decrease microvascular leakage in the gut. Importantly, in a setting of massive ischemic insult, the data suggest that the effects of PS blockade are durable [86]. This is in line with PS blockage minimizing the detrimental effects of ischemia reperfusion injury in multiple pre-clinical studies. 

The improving understanding of PS’s biology, such as the predominant PS target availability on the cell surface under pathologic circumstances, makes it an attractive target for diagnostic [34] and therapeutic interventions. Indeed, PS has been postulated to be a pharmacologic target in a number of non-ophthalmic diseases, and the therapeutic use of Annexin A5 to target PS has been proposed in RVO [11], cancer [17], sepsis [18,24], viral infections including COVID-19 [25,26], inflammatory bowel disease [27], osteoporosis [28], Alzheimer’s [13], vaso-occlusive crisis in sickle cell disease [29], diabetic skin wounds [16], and cardiovascular disease [14]. Multiple independent pre-clinical proof of concept efficacy studies related to using recombinant human Annexin A5 to target PS in vivo are available, although this is not within the scope of this review. 

## 5. The Biology and Pharmacology of Annexin A5

### 5.1. Biology

Endogenous Annexin A5 is a member of a highly conserved family of annexins (from the Greek annex, meaning “to hold together”) [87,88]. The members of the family share the same core protein structure but differ in their N-terminuses, which confer certain functional differences [88]. Annexin A5 is an intracellular protein, but also exerts its functions extracellularly [12]. 

The primary sequence of the naturally occurring human endogenous Annexin A5 protein consists of 320 amino acids. The crystal structure of endogenous Annexin A5 consists of four alpha helices with a central hydrophilic pore, while its calcium-binding sites are on the convex membrane’s binding surface [89]. Electron microscopic analysis shows close structural homology in the crystals of both soluble and membrane-bound Annexin A5, suggesting similar bioactivity of both the secreted and bound forms [90]. Atomic force microscopy shows membrane-bound endogenous Annexin A5 forming symmetrical two-dimensional network crystals, covering the cellular membrane over PS in a protective “shield”-like fashion [91]. Please refer to the recent paper by V. Gerke et al. [88], which contains the detailed information pertinent to the known aspects of Annexin A5 protein structural organization, such as the canonical annexin fold, 3D structure, as well as several diagrammatic representations.

This protein is constitutively produced, readily secreted, and acts as a paracrine-protective protein. This protein is expressed ubiquitously in human tissues, with the vascular endothelium and placenta being the major sources. Only very low extracellular levels (below 10 ng/mL) are detected in the plasma in healthy subjects. It is also present in the urine [92], amniotic [93], and cerebrospinal fluid [94].

The current state of the art indicates that the physiological role of endogenous Annexin A5 is to maintain the integrity and functionality of cellular membranes [95] and reduce cellular stress and tissue injury by dampening the amplifying role of PS, thereby limiting the initial triggering of the proinflammatory cascade and thrombogenesis. Upon the triggering of cellular stress, pre-produced intracellularly located monomeric Annexin A5 relocates immediately from the inside of the cell to the outer surfaces of cell membranes that expose PS [96]. Annexin A5 binds PS reversibly, specifically, and with high affinity to PS in the presence of physiological concentrations of Ca 2+ [97]. By building a shield over PS-exposing membranes and by neutralizing the overexposure of PS, ANXV is expected to dampen ongoing PS-dependent interactions in the downstream steps of the relevant cascades. However, during overwhelming cellular injury and hyperinflammation, characteristic of serious diseases, PS overexposure is expected to overrule the protective capacity of endogenously available Annexin A5.

Indeed, although Annexin A5 was originally identified as a protein responsible for a potent anticoagulant phenomenon in vitro [98], it is now also known to have anti-adhesive, anti-inflammatory, immunomodulating, and cell-membrane-repairing properties [21,36,99]. The importance of Annexin A5 is suggested for bone matrix remodeling [28]. Relatively little is known about Annexin A5′s biology in the eye. For example, Annexin A5 was enriched in the protein fraction derived from retinal pigment epithelium [100], whereas a recent study demonstrated that intracellular ANXA5 is required for the diurnal burst of clearance phagocytosis by the retinal pigment epithelium in vivo [101]. 

Mice lacking the Annexin A5 gene do not have an obviously altered phenotype at birth and are viable and fertile [102]. However, the maternal supply of Annexin A5 to the circulation appears to be essential for the maintenance of a fully intact pregnancy [103]. The viability of gene knock-out mice may be due to the redundancy of function of Annexin A5, whose role might be assumed according to the compensatory up-regulation and pleiotropic effects of other annexin family proteins. However, once Annexin A5-deficient mice are subjected to repeated mechanical and inflammatory stress, there is obvious damage to the heart or bone structures [104].

In disease, endogenous Annexin A5 levels have been reported to fluctuate [105,106,107]. The reduced availability of and depletion of this protein have been linked with human disease with aberrant exposure of PS, such as acute cardiovascular [106] and sickle cell disease (SCD) [76]. Cellular aging is associated with the decreased production of Annexin A5 [108], potentially making elderly patients more vulnerable to Annexin A5 depletion. 

### 5.2. Pharmacology

The majority of the pharmacologic effects of recombinant human Annexin A5 are postulated to be dependent on targeting PS on cell membranes. There is an extensive body of literature demonstrating the multiple properties of endogenous and recombinant Annexin A5 in vitro and in vivo. The pharmacologic effects of recombinant human Annexin A5 that might be of relevance to RVO and other retinal diseases include its anti-adherent [11], anti-inflammatory [18,24], anti-thrombotic [20], and cell-protecting effects [15,109]. 

Systemically injected Annexin A5 molecules behave as a “bloodhound” or a “cruise missile”, efficiently seeking throughout the body for their target, PS, on the membranes of perturbed cells [36], along with circulating microvesicles released from perturbed cells. Once bound, Annexin A5 builds a shield-like structure over membrane areas exposing PS [110]. The biological property of systemically injected Annexin A5 rapidly finding and binding its target PS on site is called “homing” [17]. 

The anti-inflammatory dose-dependent effects of Annexin A5 on vasculature were demonstrated in multiple in vivo studies, with a strong inhibitory effect on the recruitment of inflammatory cells, vascular remodeling and a reduction in fibrotic changes. The potency of Annexin A5 for reducing inflammation-driven vein-graft thickening was reported to be comparable to that of dexamethasone [19,111]. Short-term systemic Annexin A5 treatment was shown to improve endothelium-mediated dilatation [19].

Neuroprotection and the amelioration of neuroinflammation were demonstrated in a recent in vivo murine model of a traumatic brain injury with a single intravenous dose of Annexin A5. Mechanistically, it was suggested that in this model Annexin A5 acts by regulating the NF-kB/HMGB1 pathway and the Nrf2/HO-1 antioxidant system [112].

The antithrombotic effects of Annexin A5 are based on the local shielding of PS and hindering PS participation in the activation of prothrombinase complexes. Annexin A5 does not interact or inhibit the coagulation factors directly. In several non-clinical models, Annexin A5 was found to have the effects of decreasing fibrin accretion and thrombus formation and reducing platelet adhesion. Although Annexin A5 has clear dose-dependent antithrombotic activity, there is no evidence of sustained systemic anticoagulation or bleeding up to 1 mg/kg in vivo studies [20,113,114,115]. One study in a mouse model demonstrated significant bleeding in incision wounds at 2 mg/kg of rat Annexin A5 [116]. However, it is notable that there is also a substantial difference in the tertiary structure described between human and rat Annexin A5 [117], and no human recombinant Annexin A5 was included in this study [116]. 

The cell-repairing ability of Annexin A5 is linked to direct binding to PS, the resealing of membrane damage via the formation of 2D protein arrays at membrane disrupted sites, and preventing the extension of membrane ruptures. Several studies demonstrate that when recombinant human Annexin A5 is added to damaged Annexin A5-deficient cells, up to 85% of these cells are repaired within seconds [110,118] Several studies suggest that Annexin A5 protects against apoptosis and ischemia reperfusion-induced cellular damage [14,119].

Annexin A5 has been demonstrated in vitro and in vivo to contribute as a tolerogenic signal on the surface of apoptotic cells [120]. The effect of Annexin A5 on apoptotic and necrotic cells clearance has been described in vitro [121]. 

Annexin A5 was able to stimulate the release of urokinase-type plasminogen activator and the migration of rabbit corneal epithelial cells in vitro. When administered as part of an eyedrop formulation, it improved corneal wound healing [23,122]. An attempt to use Annexin A5 to facilitate drug delivery to the posterior segment of the eye was made successfully in a pre-clinical study [123]. 

Direct anti-tumor effects and the improvement of survival have been observed with systemic Annexin A5 at higher doses in several cancer in vivo models [17,124]. The potency of Annexin A5 for immune check point blockade therapy was comparable to that of other reported check point inhibitors including anti PD-1, anti PDL1, anti-TIM 3, and anti TGB-beta. This effect is proposed to be dependent on the preferential of Annexin A5 to the tumor microenvironment enriched with PS-positive tumor cells [17,24,125]. There was a clear inhibitory effect on tumor angiogenesis, and Annexin A5 was demonstrated to inhibit the secretion of VEGF in vivo [124]. 

The potential clinical benefits of Annexin A5/ANXV administration are based on the hypothesis that during overwhelming cellular injury (i.e., induced by hypoxia) and subsequent inflammation, there is no availability/sufficient protective capacity of endogenous Annexin A5 to compensate for PS overexposure, and hence the administration of exogenous Annexin A5 (ANXV) might be clinically beneficial. This hypothesis is supported by the evidence in the most extreme example of PS-dependent simultaneous multiple microvascular occlusions, namely in patients with sickle cell disease, where an imbalance between available Annexin A5 and PS expression in erythrocytes is linked to the severity of the vaso-occlusive crisis and vascular injury [30,76], with this imbalance proposed to be the driving force behind extensive vascular injury during a vaso-occlusive crisis in sickle cell disease [29,77].

## 6. ANXV (Annexin A5) as an Investigational New Drug

ANXV is an investigational new drug that contains human protein Annexin A5 (35.7 kDa) produced by recombinant techniques in *E. coli*. It has 99.7% homology with endogenous human Annexin A5. ANXV and naturally occurring human endogenous Annexin A5 proteins are both non-glycosylated and non-phosphorylated. 

ANXV has been shown in vitro to bind to PS with a binding activity and a pattern that are comparable to a reference recombinant human Annexin A5 also produced in *E. coli* [126]. The binding activity of this reference Annexin A5 has phospholipid-binding activity that is comparable to that of endogenous placenta-derived Annexin A5 [97]. ANXV is expected to have a broad therapeutic potential in diseases characterized by the increased availability of PS, where local cellular and vascular damage, thrombogenesis, and inflammation are predominant pathogenic features [19,127,128]. 

Briefly, the drug product ANXV is supplied in a clear sterile buffer solution intended for intravenous administration. Other routes of administration such as intravitreal, suprachoroidal, or subcutaneous are potentially feasible, although currently not in clinical development.

ANXV has been extensively investigated in preclinical settings, and one Phase 1 First in Human clinical trial in male healthy volunteers has been completed (NCT04850339). The first study in patient population with ANXV (Proof of Concept/Phase 2a; NCT05532735) is ongoing in the USA (recruitment completed, study is in the follow-up stage). Overall, 46 healthy male volunteers (Phase 1, First in Human clinical trial) and 16 patients with RVO (Phase 2a clinical trial) have been enrolled into the ANXV clinical program to date, of which 47 subjects have received ANXV (n = 32, healthy subjects and n = 15 patients with RVO).

In the Phase 1 First in Human clinical trial (NCT04850339), no serious adverse events (SAEs) were reported, and most adverse events (AE) were of mild intensity. No AE resulted in any action taken to the use of the investigational medicinal product ANXV. In conclusion, all investigated doses (up to 2 mg/day up to 5 days) were safe and well tolerated in healthy male volunteers. 

The overall clinical safety data from the First in Human clinical trial supported the evaluation of ANXV in the clinical trial in patients with RVO. The starting dose of ANXV was 2 mg/day/subject, for up to 5 doses administered once daily intravenously as a 30 min infusion. This dose was estimated to be within the efficacy range for RVO. Based on the emerging safety profile of ANXV and the recommendations by the Safety Review Committee, the adaptive study design of the Phase 2a study in patients with RVO allowed for dose escalations of up to 6 mg/day/subject of ANXV.

Based on the product-specific and development stage considerations, one risk mitigation strategy includes the evaluation of the immunogenicity profile of a new biologic such as ANXV. This includes, for example, the monitoring of anti-drug antibody (ADA) responses and the potential for cross-reactivity between treatment-induced ADA (if any) and endogenous Annexin A5. No evidence of pre-existing, treatment-emergent/boosted anti-drug antibodies (ADA) has been detected in healthy male subjects participating in the ANXV Phase 1 clinical trial or in Phase 2a cohorts (2 mg/day) analyzed so far.

No pharmacodynamic drug interaction studies have been performed with ANXV. Except from an experimental anti-PS monoclonal antibody bavituximab, it is not expected that other drugs would compete or interfere with the PS-binding activity of ANXV. Currently, in the early stage of development of ANXV, a very careful approach is taken and patients with RVO that receive medications that might affect hemostasis are excluded from participation in the study.

## 7. The Rationale for Investigating ANXV in Patients with Retinal Vein Occlusion

Recently, PS externalized on erythrocytes and microparticles was identified as a potential novel pharmacological target and Annexin A5 was proposed as a potential therapeutic agent in RVO [11,43,129]. Chabanel et al. reported increased propensity for the aggregation of erythrocytes derived from RVO patients [130]. Erythrocytes from RVO patients have significantly higher capacity to aggregate and adhere to the endothelium under both static and physiologic flow conditions [11], comparable to that of erythrocytes from patients with diabetes mellitus, sickle cell disease, and polycythemia vera [129]. The abnormal externalization of PS on the erythrocytes from RVO patients was linked to their significantly increased adhesiveness to the endothelium and the robustness of an erythrocyte aggregate [11]. Additionally, an overproduction of microparticles in RVO with significantly elevated externalization levels of PS was linked to an increased procoagulant capacity with shortened clotting time via the upregulation of Factor Xa and thrombin formation [43]. 

Wautier et al. investigated several PS inhibitors in an ex vivo system, where recombinant human Annexin A5 was demonstrated to be the most efficient PS blocker with up to a 70% reduction in RVO erythrocyte adhesion to microvascular endothelial cells under flow conditions [11]. 

Based on its PS binding activity, ANXV is postulated to act as an anti-adherent, anti-inflammatory, on-site anti-thrombotic (as opposed to a systemic anti-coagulant), and cell-membrane-protecting/repairing agent. Ex vivo and in vivo pharmacology studies suggest that ANXV will rapidly bind to systemically available PS-carrying erythrocytes, activated platelets, endothelial cells, and microparticles. ANXV will also accumulate at the site of occlusion and interfere with focal PS-dependent interactions, such as cell-to-cell adhesion, and potentially reduce the size of or remove the occlusive aggregate in the retinal vein, thereby limiting the retinal area of non-perfusion (RANP) and providing other short-term and long-term benefits for RVO patients. 

The anti-thrombotic and anti-fibrin accretion effects of ANXV are expected to assist in accelerating the resolution of the occlusion and reducing the risk of growth of RANP. The membrane-stabilizing/cell-protective effects may support multiple retinal cells until ischemia is relieved. ANXV is expected to provide anti-inflammatory effects on the vasculature. Murine studies with ANXV suggest that it has an effective vascular anti-inflammatory effect that is comparable to dexamethasone without immunosuppression [19]. This may complement the effects of a VEGF in inhibiting neovascularization. An additional direct effect of ANXV inhibiting VEGF secretion and angiogenesis cannot be excluded [10,124]. Short-term systemic Annexin A5 treatment in vivo improved endothelium-mediated dilatation, indicating a positive effect of Annexin A5 on nitric oxide-synthesis by the endothelium [19]. Based on the Mode of Action, ANXV is expected to be differentiated from the current standard of care anti VEGF (Appendix A).

The therapeutic window, in which near full recovery is still possible after the initial RVO event, has not been determined. It is postulated that, to maximize the potential benefit, to act as a disease-modifying agent, and potentially limit the risk of RVO becoming a chronic condition, ANXV treatment is best given at an early stage after the onset of symptoms. The ongoing clinical trial with ANXV in patients with RVO is the first clinical trial in the target patient population. This study is designed to investigate safety and tolerability and provide a proof of concept of ANXV in the treatment of subjects with recently diagnosed RVO. Enrollment was limited to RVO-treatment-naïve patients with recently diagnosed RVO and initiated ANXV treatment within 28 days post symptoms onset.

In conclusion, by targeting PS, ANXV may be effective in RVO through several mechanisms of action differentiated from, but potentially synergistic/complementary to, the effects of anti- VEGFs. 

## 8. Clinical Experience with Other Annexin A5-Related Drug Candidates

The emphasis on all available clinical experience and the safety and tolerability of systemically (intravenously) administered recombinantly produced human Annexin A5 is of the uttermost significance. We address this by explicitly describing clinical experiences with other Anenxin A5-related drug candidates, in particular the safety and tolerability aspects of these studies that might be relevant for ophthalmology.

Previously, Mui et al. reported an ongoing investigator-initiated randomized, double-blinded, placebo-controlled phase 2 clinical trial with a recombinant human Annexin A5 (SY-005) in patients with sepsis and COVID-19 [26]. The intervention was SY-005, 50 or 100 μg/kg (corresponding to a total dose of 3 and 6 mg/subject/day; assuming a 60 kg body weight), intravenously every 12 h for 7 consecutive days. With the reduction in the number of patients with severe COVID-19, the enrollment into this clinical trial was low and only 18 out of 55 eligible subjects were enrolled. There were no drug-related serious adverse events (SAEs) [31].

Prior to the study in COVID-19, a First in Human clinical trial in healthy subjects with SY-005 was reported in China (NCT042176299). This trial was randomized, double-blinded, and placebo-controlled, in which SY-005 was administered intravenously. The SAD part of the trial investigated dose levels from 0.75 to 20 mg/subject in six escalation steps, while the MAD part investigated 5 mg, 10 mg, and 20 mg/subject/day administered for 7 consecutive days. The clinical trial report made available by the sponsor on the open access database states that no subjects had SAEs, and no withdrawals from the study or deaths were reported [131].

There is favorable clinical safety experience in relation to intravenously administered minimally modified recombinant human Annexin A5 labelled with a radioligand (Tc99m) or a fluorescein dye that retains the binding abilities of the recombinant not labelled as Annexin A5. The purposes of the labelled Annexin A5 are the diagnosis and assessment of the response to treatment in the dose range of a single bolus dose of 0.025 to 1 mg (0.0004–0.0166 mg/kg). Among all of the published clinical studies performed with recombinant human Annexin A5 derivates, none have reported signs of allergic or anaphylactic reactions. Altogether, 76 healthy subjects and 358 patients have been included in these studies [33]. 

Labelled recombinant human Annexin A5 is reported to be eliminated primarily in the urine and a minor fraction in the feces in humans [36,132]. In addition, the splenic reticuloendothelial system is expected to clear ANXV-carrying microparticles from circulation. 

Other approaches targeting PS have been reported without reported untoward side effects related to the PS targeting. There is also favorable safety experience in renal transplant patients administered single IV bolus doses of up to 24 mg (0.4 mg/kg considering 60 kg body weight) of a recombinant homodimer of endogenous human Annexin A5 (Diannexin) [133]. 

In addition, there are long-term clinical safety and tolerability data related to a fully human monoclonal IgG1 antibody (bavituximab) targeting the PS-binding plasma protein beta2Gp1 at dose levels of up to 3 mg/kg administered for several months [134]. Bavituximab has shown some success in Phase 2 trials and is currently being explored in combination with check-point inhibitors in several difficult-to-treat human cancers. These programs have, to the best of the authors’ knowledge, not reported adverse events related to the targeting of PS. 

There is thus ample clinical experience relating to single-dose administrations of modified Annexin A5 products, with expected biological functions comparable to those of a non-modified Annexin A5, at dose levels significantly above those proposed/anticipated for efficacious doses of ANXV, as well as experience with the long-term targeting of PS in the clinical setting through the use of a mAb. 

## 9. PS Binding Annexin A5-Based Molecular Diagnostic Tools

Based on its ability to identify cells that are undergoing apoptosis or transiently exposing PS, there is potential for Annexin A5 proteins not only to identify and quantify the degree of degeneration or vascular damage in the retina but also to identify patient groups more likely to respond to a drug that blocks PS signaling, as well as to follow the effect of a drug attempting to protect the retina from degeneration. Provided success with PS-targeting therapeutics and diagnostics, a theranostic approach to optimize ophthalmic therapies can be envisioned. 

Indeed, diagnostic tools based on PS-binding Annexin A5 proteins were developed in the 1990s and early 2000s, with Prof Blankenberg at Stanford, CA and Prof Chris Reutelingsperger, Univ of Maastricht, The Netherlands, as pioneers. Variants of Annexin A5 proteins aimed to facilitate the chemical binding of short-lived radioisotopes, mainly technetium-99m (99mTc), which served as diagnostic candidates in several clinical trials. Annexin A5 radioligands were explored in multiple medical conditions including cancer, bacterial endocarditis, arthritis, chronic obstructive pulmonary disease, and carotid artery disease [135,136]. Their overall clinical utility as single-photon emission computed tomography (SPECT) ligands was proposed to be as high as, e.g., 18F (fluorodeoxyglucose) in detecting metastatic cancer [35]. No safety or tolerability issues were reported in studies of approximately 400 patients who received different variants of radiolabeled Annexin A5s [35]. While the emerging data were promising, the clinical development did not move forward to make this a commercially available diagnostic tool. To the authors’ knowledge, the usefulness of positron emission tomography (PET) ligands of Annexin A5 has been demonstrated in animals but has not yet moved forward to the clinic. Possibly, and with PET technology being superior in terms of image resolution and PET cameras becoming more readily available, radioligands supportive of PET technology represent an attractive opportunity, potentially also in retinal disease.

In retinal disease, a technology abbreviated as DARC (detection of apoptosing retinal cells) has been developed by Dr Cordeiro and co-workers since the beginning of the 2000s [34]. It is currently made available and in use in the support of clinical drug development with retinal cell sparing drugs. With DARC, a fluorescently labelled Annexin A5 (ANX776) is used in combination with improved machine-learning imaging algorithms to detect apoptotic retinal ganglion cells, found to show sensitivity at the single-cell level. The clinical safety and tolerability of ANX776 in a total of 129 patients, of which 113 were patients, were deemed to be favorable [34]. The utility of ANX776 as a diagnostic tool in retinal disease has been explored in diabetes, glaucoma, and AMD. Pending further validation, ANX776 may become an important tool in the diagnosis, patient selection, and treatment follow-up in multiple retinal diseases.

## 10. Conclusions

Vascular retinal disorders lack easy-to-administer drugs with an effect on the causative pathophysiology, and are limited to addressing the consequences, e.g., retinal edema, neo-angiogenesis and inflammation, rather than addressing the cause. A comprehensive understanding of the key molecular players involved in the retinal microenvironment subject to alterations by the underlying disease continues to be important. Potentially, a disease-modifying impact will ultimately allow for designing and exploring novel safe therapeutic or combination therapies approaches to further mitigate visual loss.

New mechanistic insights into the pathogenesis of retinal diseases with a prominent vascular component such as RVO, DR, and wet AMD may open up new paradigms of treatment that reach beyond the inhibition of VEGF. PS is a potential novel lipid target that arguably can be a key driver of pathology in vascular retinal disease and is targetable by Annexin A5 proteins. The investigation of PS inhibition by systemically administered ANXV (recombinantly produced human Annexin A5) in patients with RVO is one example of efforts to provide additional safe and effective therapies in the retinal disease space.

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
