# Peer review of "New Therapeutic Strategies in Retinal Vascular Diseases: A Lipid Target, Phosphatidylserine, and Annexin A5—A Future Theranostic Pairing in Ophthalmology"

_pharmaceuticals, 2024, doi:10.3390/ph17080979_

Round 1

Reviewer 1 Report

Comments and Suggestions for Authors

Point 1:- First of all how come abstract contains references, (line 13,17,20,24 and many more

Point 2:- abstract should be within word limit not more than journal instruction 

Point 3:-  Modify whole abstract and shorten it within word limit without references 

Point4:- line74-77, what exactly the sentences was going to tell. confused with sentences meaning

Point 5:- authors contribution not given , confict of interest , try to follow journal instruction properly

Point 6:-  Phosphatidylserine and Annexin A5 are used in ophthalmic disease , but try to explain more with diagrammatic way how they can resolve the disease.

Point  7:- actually what was the exact theme of the article review, Phosphatidylserine is cargo of the drug and Annexin A5 is therapeutics drug and how it is helpfull as in which formulation , give more depth on it 

Point 8:- explain the clinical trials in depth along with tables with different researcher had given different input

Point 9:- focus on the structure of the sequences of writing, 

Point 10:- More emphasis on formulation concept with theranostic property should be given

Author Response

Please see the pdf attachment.

The Academic Editor's Report and the English Editing Certificate are attached as Supplementary Files.

Reviewer 2 Report

Comments and Suggestions for Authors

Please find the document attached.

Author Response

(The authors gave the same response as above.)

Reviewer 3 Report

Comments and Suggestions for Authors

Dear Author,

Your manuscript entitled “A Lipid Target Phosphatidylserine and Annexin A5–A Future Theranostic Pairing in Ophthalmology” has been thoroughly investigated. After a comprehensive evaluation process, the manuscript needs to be heavily revised. After point-by-point fulfillment of requested revisions, the manuscript could be a candidate for a review study. Hereby, I would like to present my comments:

1. Please consider rewriting the abstract part. The abstracts should be more concise and should not repetitive. Please examine and analyze the previous abstracts from the published materials in the journal.

2. The grammar and structure of the manuscript should be monitored. A professional editing service should be more beneficial in this context. Please consider getting editing service of the journal.

3. (As a technical note, Lines 90-91: citations needed after the sentence): “…Treatment with Annexin A5 counteracts downstream effects of PS externalization as demonstrated in multiple pre-clinical studies…”. Please consider citing this sentence with an appropriate reference.

4. (As a technical note, some of the typos should be revised, Line 96, Line 100, Line 179): “…cell membrane repair agent…, …. in sickel cell disease…, … floppase…”. Please consider a thorough investigation throughout the text for elimination of these kinds of issues.

5. Please consider simplify or clarify some of the sentences for better understanding of your expressions. (e.g. “Despite of the excellent efficacy of anti-VEGFs in reducing macular edema and subsequent rapid improvement of visual acuity, in many patients the efficacy tends to decline when injection frequency is reduced, while in others persistent macular edema is obvious despite intensive VEGF treatment”). Please consider a thorough investigation throughout the text for elimination of these kinds of issues.

6. Please check the whole text for the better explanation of abbreviations (IVT injections, derived from RPE, COPD and carotid artery disease, PET ligands…etc.)

7. Some of the paragraphs or expressions should be removed or clearly correlated with the topic throughout the text. The relevance of the paragraph should be explained in between the lines 240-245. The section subheading was “PS as a Novel Therapeutic Target in Ophthalmic Diseases”, but the paragraph mentions about cancer or tumor microenvironment.

8. (As a technical note, Lines 343-348: citations needed): “The majority of the pharmacologic effects of recombinant human Annexin A5 are postulated to be dependent on targeting PS on cell membranes. There is extensive literature demonstrating multiple properties of endogenous and recombinant Annexin A5 in vitro and in vivo preclinical studies. The pharmacologic effects of recombinant human Annexin A5 that might be of relevance in RVO and other retinal diseases are anti-adherent, anti-inflammatory, anti-thrombotic and cell protecting.”

9. As a major structural request, the boundary of the review needs to be demonstrated. What was the inclusion or exclusion criteria? What was the rationale of subheading or subtopic selection. As a critical example: systematic reviews use PRISMA approach (https://doi.org/10.1136/bmj.n71) for better understanding of main or subtopics of the intended research area. Please make an informative introduction part, then share these critical points for your review study.

10. Please indicate your review’s boundary or exclude whole paragraph in lines 421-424: “ANXV is currently in clinical development by Annexin Pharmaceuticals (Stockholm, Sweden). The ANXV drug product related information, non-clinical safety and toxicology, pharmacokinetics as well as the Phase 1 detailed data and the dose rationale for Phase 2a in RVO are outside the scope of this review and will be made available elsewhere.”

11. Please cite the clinical trial no or NTC code in lines436-440: “In Phase 1 First in Human clinical trial no serious adverse events (SAEs) were re-436 ported, and most adverse events (AE) were of mild intensity. No AE resulted in any action 437 taken to the use of investigational medicinal product ANXV. In conclusion, all investi-438 gated doses (up to 2 mg/day up to 5 days) were safe and well tolerated in healthy male 439 volunteers.”.

12. Please explain the relevance of the section “7. Clinical Experience with other Annexin A5-Related Drug Candidates”. The review study is about the Annexin, PS targeting, theranostics and ophthalmology. Thus, there were two or three options for this part: i)make an informative starting paragraph to connect this part to main topic, ii)make an informative introduction part to demonstrate your boundaries of your review study, (iii)exclude this section to clarify the other issue (the subject should be focused on key points).

13. The reference format needs to be checked and revised. Please reconsider following the instructions for authors (referencing style must be revised, also please use the journal abbreviations in the references section (REF17, 74, 100, 101, 105, 106, 110, 116, 117, 120, 121, 127, 129, 132.)

Best regards.

Comments on the Quality of English Language

Dear Author,

Upon reviewing your manuscript, it was found to contain numerous typos and grammatical errors. Additionally, the structural length of some paragraphs, as well as the use of conjunctions and transitional phrases, may make it challenging for readers to follow. Considering all these points, I believe it would be beneficial for you to utilize the journal's editing service.

Sincerely,

Author Response

(The authors gave the same response as above.)

Round 2

Reviewer 1 Report

Comments and Suggestions for Authors

point 1:- all comments had fulfilled and corrected

point 2:- author contribution not given in end 

Reviewer 2 Report

Comments and Suggestions for Authors

Answers to questions and comments are convincing.

Reviewer 3 Report

Comments and Suggestions for Authors

Dear Author,

Although the draft of the manuscript you uploaded is not suitable, the previous revision process has been evaluated based on the Word file you uploaded in the non-published material section of the system. By utilizing the editing service provided by the publisher and examining all other sections, it has been observed that the previous requests have been successfully addressed. However, a minor revision has been given to draw the editor's attention in terms of the publication process.

Respectfully submitted for your information.